# Anxiety Levels and Coping Strategies to Deal with COVID-19: A Cross-Cultural Study among the Spanish and Latin American Healthcare Populations

**DOI:** 10.3390/healthcare11060844

**Published:** 2023-03-13

**Authors:** María Angustias Olivencia-Carrión, María Demelza Olivencia-Carrión, Martha Fernández-Daza, Sara Zabarain-Cogollo, Greys Patricia Castro, Manuel Gabriel Jiménez-Torres

**Affiliations:** 1Health Psychology/Behavioural Medicine Research Group (CTS-267), Universidad de Granada, 18071 Granada, Spain; 2Torredonjimeno Health Centre, 23650 Jaén, Spain; 3Psychology Program, Universidad Cooperativa de Colombia, Santa Marta 110000, Colombia; 4Interdisciplinary Social Studies Research Group-ESI, Santa Marta 110000, Colombia; 5Department of Personality, Evaluation and Psychological Treatment, Faculty of Psychology, Universidad de Granada, 18071 Granada, Spain

**Keywords:** anxiety, coping strategies, COVID-19, pandemic, healthcare population

## Abstract

Given its impact, COVID-19 has engendered great challenges in terms of health, highlighting the key role of health personnel. This study aims to analyze the level of anxiety, as well as coping strategies, among the health personnel in Latin American countries and Spain. An exploratory, descriptive, quantitative, cross-sectional study was conducted with 584 participants from the healthcare population. No significant differences were observed in anxiety levels due to COVID-19 between Latin American countries and Spain. In Spain, an active and passive coping style is used, while in Latin American countries, an avoidance coping style is employed; there is a direct correlation between anxiety levels and the avoidance coping style. There exists an inverse correlation between anxiety levels and the use of an active coping style; moreover, there are no significant differences in the anxiety level of health personnel depending on whether they have cared for patients with COVID-19. Low cognitive activity, use of the avoidance method and Spanish geography were the main predictive coping styles of anxiety. Effective measures are required for preserving the mental health of health professionals during pandemics.

## 1. Introduction

The COVID-19 pandemic has led to a major global public health crisis. It is an unprecedented event in the 21st century and has posed a comprehensive challenge for everyone due to its impact on and consequences in different areas of life, such as health, the economy, and society in general [1,2,3,4]. As per [1,5,6,7,8,9] the COVID-19 health crisis has had a global impact in a short period of time, given that health professionals went from a stable situation, to living in a work environment characterized by overcrowded hospitals, lack of personal protection equipment, nonexistent or contradictory work protocols, and an increase in mortality.

Additionally, health professionals have been directly involved with their COVID-19 patients at different stages—diagnosis, treatment, care, and in the process of death—as family members were unable to accompany COVID-19 patients, who ended up dying alone. Because of this, they are more vulnerable to psychological disorders, such as anxiety [2,5,6,7,8,9,10,11,12,13,14,15,16,17,18,19,20,21,22,23,24,25,26,27].

Anxiety is defined as a set of physical, mental, and motor manifestations, not attributable to real danger, but arising suddenly or as a constant and imprecise state [17,18,25,28,29,30,31].

Another important aspect is the coping strategies that individuals utilize during critical events. Ref. [32] suggested that coping is a changing process for adaptation purposes, in which the subject and the environment constantly interact, and that people modify their coping strategies depending on the type of problem to be solved. Likewise, coping strategies refer to a set of responses that a person puts into practice to resolve challenging situations and reduce the tensions generated by them [28,33]. Therefore, coping has been shown to be a key factor for psychological well-being and mental health problems during the COVID-19 pandemic [17,21,25,33,34,35,36].

Regarding coping strategies and their relationship with anxiety, the ones most commonly used by people with anxiety are problem-solving and positive reappraisal. The cognitive avoidance strategy is also used, which is based on trying to withdraw or flee from the stressor and its consequences [37]; it is characterized by the use of passive strategies that consist of avoidance behaviors [38].

Several studies have elaborated the possible coping strategies associated with anxiety [21,25,28,31,33,34,35,39,40,41,42,43,44,45,46,47,48].

In the general population, there are several factors related to psychological impact, such as a lower perceived state of health, being female, a highly perceived vulnerability to COVID-19, and exposure to negative news [3,33,34,35,49]. These factors are linked to the perception of less social support and resilience [21,29,50,51,52,53,54,55,56].

Ref. [49] showed that the COVID-19 preoccupations were significantly and positively correlated with anxiety, depression, and stress severity, and increased the use of maladaptive copings. Furthermore, their findings highlighted that the various stresses of the 21st century due to the COVID-19 pandemic are related to personal well-being, thus it is important to consider the potential global mental health in general population.

Specifically, the healthcare population has had to fight with an unknown virus, unsafe measures and constant work stress, where deaths increased at an unbridled rate without the ability to find safe treatments [2,4,6,7,8,9,11,14,15,18,23,24,34,39,40,41,42,45,57,58,59,60,61,62,63,64,65].

Moreover, the psychological impact produced by the COVID-19 pandemic can manifest itself as serious problems involving a deterioration in the health and operation of the healthcare population, resulting in anxiety, sleep disorders, stress, depression, burnout, acute stress, and post-traumatic stress disorder [1,2,4,5,6,7,9,10,11,12,13,14,16,17,19,20,23,24,26,29,35,39,40,41,42,45,48,52,60,64,66,67,68].Various studies have shown that the presence of anxiety is associated with symptoms of stress, nervousness, fear, fatigue, viral symptoms, job disappointment, perception of discrimination, and negative actions related to COVID-19 [1,5,6,7,9,10,11,19,20,22,23,24,36,59,62,63,69].

Regarding Spanish studies that dealt with anxiety levels and coping strategies in health professionals, we found the following: Ref. [29] showed that being a health professional, having a lower level of perceived subjective stress, and a greater proportion of active coping were significant predictors of fewer psychopathological symptoms. Ref. [13] highlighted that the high scores for resilience were significantly associated with a better quality of life and lower levels of anxiety, depression, and post-traumatic stress in the healthcare population. Ref. [59] revealed that anxiety levels were higher in women and that working with patients with COVID-19 increased their anxiety levels. Ref. [70] emphasized the critical value of mental health professionals during the early stages of the pandemic for those caring for patients with COVID-19. Ref. [71] analyzed the anxiety levels of health professionals derived from the death of their patients, and found that the lack of personal protection equipment and exposure to death resulted in high levels of anxiety. Ref. [53] examined anxiety and resilience in health professionals, showing negative correlations between resilience and symptoms of generalized anxiety disorder, which was more prevalent among women.

Regarding Latin American studies related to anxiety and coping strategies in the healthcare population, we found the following:

In Peru, Refs. [11,14] showed the relationship between fear of contagion and physical–cognitive fatigue, and a significant relationship between generalized anxiety and physical–cognitive fatigue. Ref. [69] revealed that female health workers in particular, such as nurses and those who work directly with COVID-19 cases, dealt with anxiety. Further, Ref. [68] obtained similar results. Additionally, Ref. [67] determined the relationship between the high level of generalized anxiety disorder and health personnel who provided care for patients with COVID-19.

In Argentina, Ref. [5] revealed that the factors most related to the presence of psychic distress are: direct work with patients with COVID-19, female gender, younger age and nursing profession. Ref. [25] examined the relationship between anxiety and coping in the context of the COVID-19 pandemic in the healthcare population.

In Colombia, Ref. [12] found that there was a decline in the sleep quality of mental health workers during the COVID-19 pandemic. Ref. [72] highlighted that with physicians during the COVID-19 quarantine, detecting anxiety symptoms among health personnel attending patients infected with COVID-19 was a current priority. Ref. [15] revealed a higher prevalence of post-traumatic stress disorder, anxiety, and depressive symptoms in healthcare workers during the COVID-19 pandemic in Colombia. Ref. [62] revealed the relationship between generalized anxiety disorders and the fear of COVID-19 in doctors during the pandemic, associating other symptoms of anxiety, such as stress, fear, job disappointment, and perception of discrimination. In a previous study, anxiety related to the pandemic and professional work was found to be highly prevalent and more frequent among physicians with perceived discrimination [63].

In Mexico, Ref. [73] showed that the increase in psychological stress and work overload was associated with the appearance of a generalized anxiety disorder, among other factors. Refs. [22,23] revealed higher anxiety scores among health professionals who had coronophobia, a higher perception of risk of contracting COVID-19 or infecting family members, greater uncertainty as to how to properly cope with the pandemic and those working in emergency rooms with patients with COVID-19.

In Ecuador, Refs. [20,36,54] showed that the healthcare population presented psychological distress and used positive coping strategies to continue with their work.

In Brazil, Ref. [7] revealed that the prevalence of anxiety, depression and stress symptoms in the health personnel indicated a high risk of mental illness in health professionals during the COVID-19 pandemic.

Finally, in Latin America, recent studies have determined that healthcare workers suffered middle-higher acute stress due to the outbreak, and experienced acute stress, increasing in intensity as the incidence of COVID-19 increased [19]. A previous study proposed to disseminate resources to mitigate the effects of COVID-19 and reflect on the role of psychologists as part of the healthcare population during the pandemic [74].

The general aim of this study was to analyze the anxiety levels and coping strategies among health personnel in Latin American countries and Spain. The specific study objectives are as follows:Comparing the levels of anxiety due to COVID-19 among health personnel in Latin American countries and Spain;Checking the differences by sex in the levels of anxiety due to COVID-19 among health personnel;Comparing the strategies for coping with COVID-19 by health personnel;Determining the influence of coping strategies on the level of health personnel anxiety because of COVID-19;Observing the difference in anxiety levels between health personnel who have cared for patients diagnosed with COVID-19 and those who did not;Studying the predictive factors of anxiety due to COVID-19 among health personnel.

The main question addressed by the research is: Does the type of coping strategy used by the healthcare population to deal with COVID-19 have an impact on the level of anxiety experienced by health professionals?

## 2. Materials and Methods

### 2.1. Participants

The sample comprised 584 Hispanic American health professionals, who worked during the COVID-19 pandemic and who participated in this study voluntarily. This was an incidental sampling drawn from the Spanish and Latin American healthcare populations. The age range was between 18 and 75 years, and participants had different personal and sociodemographic conditions (Table 1). It was a non-probabilistic convenience sample. The sample size was estimated before the study, using the online calculator of the Clinical and Translational Science Institute (University of California, San Francisco, CA, USA) for clinical correlational research [75].

### 2.2. Instruments

The following measures have been used for data collection: the Zung Anxiety Scale [76], Spanish version [77], which is a self-report measure comprising 20 items, designed according to a Likert-type scale of four points (1–4) that measures anxiety symptoms. It presents a minimum score of 20 and a maximum of 80, with the highest values corresponding to more anxiety. Items 5, 9, 13, 17, and 19 are written in reverse order (no symptoms or anxiety). These items are distributed in four anxiety subscales: Cognitive (Items 1–5), Motor (Items 6–9), Vegetative (Items 10–18) and Central Nervous System Anxiety (Items 19 and 20). Results greater than or equal to 40 are values indicative of pathology. This scale has good validity, reliability, and discrimination.

Total raw scores range from 20 to 80 points. This raw score is converted to an anxiety index, using a conversion table. According to this anxiety index, four levels of anxiety are differentiated: no anxiety (20–44 points), mild anxiety (45–59 points), moderate/severe anxiety (60–74 points), and extreme anxiety (≥75 points). This scale has well-established psychometric properties [78,79]. For the present study, a Cronbach’s α value of 0.95 was obtained.

Scale of Styles and Strategies for Coping with Stress [38] is a 72-item self-report that assesses 18 different strategies: positive reappraisal, depressive reaction, denial, planning, conforming, cognitive disengagement, personal development, emotional control, detachment, suppression of distracting activities, restrain coping, avoidance coping, problem-solving, social support for the problem, behavioral disconnection, emotional expression, emotional social support and palliative response, and eight different coping styles depending on the methods, approach, and type of activity used: active, passive, and avoidance coping; response-, problem-, and emotion-focused coping; and behavioral and cognitive coping (Table 2). Subjects responded to each item using a Likert-type scale ranging from 0 (never) to 3 (always). The higher the score obtained, the more commonly the coping strategy was used. Then, the coping styles were scored by adding the scores obtained in the corresponding coping strategies, noting that each strategy style is equal to that of the coping strategies. Ref. [80] reported a Cronbach’s α of 0.73 for the full scale and 0.83 for styles. In the present study, α for the full scale and styles were 0.93 and 0.96, respectively.

Additionally, the participants answered questions about their sociodemographic and personal data (age, sex/gender, marital status, educational level, employment status, number of children, and number of cohabitants). Finally, they indicated whether they had had symptoms of COVID-19, had been diagnosed with COVID-19 through a positive test, or have not had symptoms or a disease diagnosis.

### 2.3. Procedure

The researchers distributed an online survey in a single document to health personnel in the participating countries. The study was open from May 2020 to February 2021. This survey brought together different sociodemographic data, questions related to COVID-19, and the two aforementioned instruments. Participants were requested to answer the survey and send it back. All measurements were completed in a single application by the participants. Finally, the database was downloaded and verified. Participants who did not meet the inclusion criteria (i.e., age 18 or older, read and write Spanish fluently, and voluntary participation) or exclusion criteria (i.e., serious physical or mental health problems) were removed from the analysis. As all questions in the survey were mandatory, there were no participants with missing or incomplete data that had to be removed from the analyses.

### 2.4. Data Analysis

A non-experimental, cross-sectional, descriptive, correlational design was used [81]. The statistical treatment of the data was conducted with the SPSS 25.0 program (Statistical Package for Social Science, 2008). To verify the accuracy of the data entered and to know their characteristics, preliminary and exploratory analyses were carried out. Since a Levene’s test confirmed the homogeneity of variances (*p* > 0.05), we decided to use parametric tests in the statistical analyses. The level of significance for all analyses was set at *p* < 0.05 (bilateral). Descriptive analyses and independent sample comparisons using a Student’s *t*-test, Pearson’s bivariate correlations, and stepwise multiple linear regression analysis were performed.

## 3. Results

### 3.1. Comparison of Anxiety Levels due to COVID-19 among Health Personnel in Latin American Countries and Spain

As can be seen in Table 3, there were no significant differences in anxiety levels due to COVID-19 between Latin American countries and Spain (except in cognitive symptoms of anxiety, which were higher in the Spanish population).

### 3.2. Differences by Sex in the Levels of Anxiety due to COVID-19 among Health Personnel

Women showed higher levels of anxiety during COVID-19 than men (Table 4).

### 3.3. Comparison of Strategies for Coping with COVID-19 by Health Personnel

As can be seen in Table 5, a passive coping style was used more frequently in Spain than in Latin American countries. In these countries, an avoidance coping style was more commonly used.

### 3.4. Influence of Coping Strategies on the Level of Anxiety in Health Personnel (Correlations between Anxiety and Coping)

As shown in Table 6, a direct correlation was found between the level of anxiety and the use of an avoidance coping style (0.152 **). This coping style is significantly associated with both somatic and cognitive symptoms of anxiety (0.146 ** and 0.134 **, respectively).

In contrast, an inverse correlation was found between the level of anxiety and the use of an active coping style (−0.122 **).

### 3.5. Differences in Anxiety between Health Personnel Who Have Cared for Patients Diagnosed with COVID-19 and Those Who Did Not

No differences were found in the level of anxiety of health personnel, whether or not they cared for patients with COVID-19 (Table 7).

### 3.6. Predictive Factors of Anxiety due to COVID-19 among Health Personnel

The avoidance method, low cognitive activity and location (Spain) were the main predictors of anxiety (Table 8). The following factors were excluded: sex, having treated COVID-19 patients, and other coping strategies.

## 4. Discussion

This study aimed to analyze the level of anxiety and coping strategies among health personnel in Latin American countries and Spain, and the specific goals were the following: to compare the levels of anxiety due to COVID-19 among health personnel in Latin American countries and Spain and to study the predictive factors of anxiety due to COVID-19 among health personnel in Latin America and Spain, among others.

The study results indicated that there are no significant differences between the levels of anxiety due to COVID-19 between Latin American countries and Spain among health personnel. This seems to indicate that health professionals tried to protect their psychological health, and therefore their levels of anxiety, in the face of such an unprecedented threat in our recent history. These results support other studies that revealed that positive mental health and social, emotional, and psychological well-being have a positive effect, confer resilience, reduce the negative consequences of unpleasant experiences, promote an adaptive response to uncertain situations, and reduce the risk of suicidal ideation [13,14,24,25,29,30,31,34,36,40,45,49,53,54,82,83,84,85].

Thus, Refs. [2,5,6,7,9,11,22,23,26,58,60] contradict our findings, as they showed that the main sources of anxiety among health professionals were due to patient care, concern about becoming infected or infecting family members, work-related concerns, burnout, and fear of the unknown. Our results are inconsistent with [2,5,6,11,16,19,20,22,23,26,36], that reflected that the COVID-19 pandemic has generated higher levels of anxiety among health workers, regarding factors such as having been in contact with the virus or fear at work. Moreover, our results are contrary to the studies by [18,60], as they showed that health professionals have had to develop their profession in a precarious environment, putting both their individual and collective health at risk, considerably increasing their patients’ death anxiety; the predictor variables of this anxiety are the absence of personal protective equipment and high levels of burnout, emotional exhaustion, and depersonalization.

However, the findings of [52] support ours, as they showed that the prevalence of anxiety was similar between health workers and the general population.

Other Latin American studies contradict our results, as they narrate the effects of the COVID-19 pandemic on mental health found in health personnel and those who work directly with suspected or confirmed cases of COVID-19 [5,7,11,15,19,22,23,36,69]. Additionally, other studies that differs from our results is that of [35,67], which consider that anxiety shows a steady increase from the beginning of the pandemic to the present among the healthcare population. This could be because the pandemic itself is an anxiety-generating agent.

In contrast, our results show that women present higher levels of anxiety regarding COVID-19 than men among health professionals, and we found significant differences based on gender. Our results are consistent with those of [2,5,6,7,11,13,15,20,36,59,69], given that female health professionals present greater anxiety regarding COVID-19 compared to male health professionals. This could be due to the caregiver role that women play in their homes, thus resulting in greater anxiety due to the fear of contagion. However, the results should be contextualized, because a little over 80% of the sample in this study were women, with professions linked to gender (nurses and nursing assistants). Moreover, our findings are supported by studies by [2,3,6,11,15,33,39,40,42,45,52,62,63,64,68,86], as they show that being a woman is decisive in exhibiting high anxiety about the consequences of COVID-19, regardless of the country of origin of the study.

When comparing coping styles between Spain and Latin American countries, Spain used the passive coping style more, while in Latin American countries, health professionals used the avoidance coping style more. From Spain, our results are inconsistent with study by [29], which showed that being a health professional, having a greater proportion of active coping and lesser passive coping were significant predictors of fewer anxiety symptoms. These data also differ from a Spanish study by [43] of the general population, in which avoidance coping was one of the main predictors of anxiety levels during the pandemic. This shows that these strategies may reflect ineffective ways of coping, because problem-solving and a perspective change could be a valid approach for moderate anxiety symptoms. In accordance with [17,87], we examined the link between different coping strategies with anxiety symptoms and quality of life, both cross-sectionally and longitudinally, finding that avoidance coping was associated with greater anxiety and lower quality of life at the start of the study, thus supporting our findings in the present study.

Our findings reflect that there is a direct correlation between the level of anxiety and the avoidance coping style associated with somatic and cognitive symptoms of anxiety. Our data support studies on the healthcare population. Health professionals perceived that their psychological health worsened during the COVID-19 pandemic, and higher levels of anxiety were associated with increased avoidance coping strategies [7,14,17,41,45,50,57]. Thus, people with high levels of anxiety symptoms were prone to a maladaptive response to uncertain new situations [1,9,16,17,35,39,40,64].

However, there are studies that used more aggressive reaction strategies, expressions of coping difficulty and sought professional support and emotional avoidance [1,2,5,7,9,11,12,17,19,23,28,33]. This may be due to more mental health problems, fear of getting sick and infecting their families, fear of death, regret for being a health professional, not wanting to face reality, less time to rest, more night shifts, having a family, a lack of confidence in the fight against the virus, lack of training on pandemic protection, and an inadequate professional attitude.

Furthermore, our results show an inverse correlation between anxiety levels and the use of an active coping style. Studies that contradict our results indicate that at a higher level of anxiety, there exists less use of adaptive coping strategies and, consequently, greater use of maladaptive coping strategies, both in the healthcare and in general population [21,25,28,29,30,33,34,35,40,49,50,51]. This could be justified given that people with the self-perception of being healthier adopted more positive coping strategies, such as emotional coping, behavioral coping, and social support, and therefore suffered less anxiety than those with poorer self-perceived health.

Our results show that there are no significant differences between the anxiety levels of health personnel depending on whether they provided care for patients with COVID-19. These results differ from those of [11,67], and other authors who showed that there were mental health problems, such as anxiety, depression, post-traumatic stress in the healthcare population, mainly in people working in the front line with COVID-19 patients, which may be due to the fear of contagion, work stress, and witnessing coworkers die [1,5,6,7,8,18,19,22,23,26,39,40,41,42,44,45,52,53,57,58,59,61,62,63,65,69,86].

Finally, our results show that the avoidance method, low cognitive activity and location (Spain) were the main predictors of anxiety. These findings refute the conclusions of [43] that avoidance coping was one of the main predictors of anxiety levels in the general population during the pandemic. Thus, Ref. [33] study, conducted with a university population, differs from ours in term of the results, as the students expressed high levels of coping strategies and comparatively low levels of anxiety. Additionally, cognitive coping, emotional coping, and social support proved to have a significantly negative predictive effect on anxiety, with social support being the most powerful predictor. These results support [6,51] studies, which found that social support was negatively correlated with the level of anxiety in university students and health professionals, respectively. Social support not only reduces psychological stress during pandemics, but also changes attitudes regarding help-seeking methods. This result suggests that strong and effective social support is needed during a health crisis.

### Study Limitations

The present study has some limitations. First, it was based on an online self-reported questionnaire (April 2020–January 2021), the hardest period in terms of psychological impact, which may influence the data gathered. Another limitation is the lack of literature on comparisons of the healthcare population in Latin America and Spain and their relationships in terms of anxiety levels and coping strategies. Most of the studies consulted are based in Asia, the United States, and Europe, and focus more on showing the pandemic situation by country due to COVID-19, making it difficult to endorse or contradict our results. Furthermore, we have faced serious difficulties in finding such specific studies that support or refute our findings, due to the limited literature on health personnel and their relationship with anxiety and coping strategies during the COVID-19 pandemic; however, there is more literature on the general population.

In contrast, it is positively highlighted that as far as we could ascertain, our study is the first work to be currently carried out based on analyzing the level of anxiety and its relationship with coping strategies among the healthcare populations of Spain and Latin America.

Regarding future implications for clinical practice and research, it should be emphasized that health professionals are directly involved with patients with COVID-19 in different phases, such as diagnosis, treatment, care, and even in the process of death. As a result of this whole situation, they are more vulnerable to psychological disorders, such as anxiety; therefore, investigating the relationship between anxiety levels and coping strategies in the healthcare population should continue. It is recommended to conduct different follow-up assessments of health personnel from 6 to 12 months to analyze the differences during the several critical periods of the duration of the COVID-19 pandemic.

## 5. Conclusions

Coping styles and strategies influence the level of anxiety experienced in the healthcare population due to the COVID-19 pandemic. In conclusion, it is essential to reduce the psychological impact on health personnel. To do this, specific training on COVID-19 is recommended, such as reinforcing security measures, guaranteeing their basic needs, practicing adequate coping strategies to reduce anxiety, providing greater emotional support networks, and continuing research on which predictive factors contribute to a greater well-being in health promotion and anxiety prevention.

The pandemic nature of COVID-19 makes this a relevant and novel study (unprecedented). The study is relevant because it provides some knowledge of this recent COVID-19 pandemic among the healthcare population. Therefore, its results can be extrapolated to other possible pandemics that may occur in the future. It is important to disseminate the information of our results among the healthcare population, in order to reduce their anxiety levels and to encourage the use of appropriate coping strategies in future pandemics.

This underlines the need to increase our understanding of the psychological needs of individuals in the 21st century, in which people are under pressure from various problems that impact their quality of life.

Understanding these coping strategies provides insight into areas that need to be addressed, to build and maintain a workforce within the healthcare system.

More studies such as ours are needed to better describe coping strategies and styles among healthcare workers, as they relate to anxiety during the COVID-19 pandemic, as well as identify effective resources that support the psychological well-being of healthcare workers.

## Figures and Tables

**Table 1 healthcare-11-00844-t001:** Sociodemographic data of the participants.

Variables		*N*	%
Sex	Women	424	72.6
	Men	160	27.4
Age	18–44	410	70.2
	45–59	136	23.3
	60–74	37	6.3
	75–90	1	0.2
Marital status	Married	197	33.7
	Divorced/separated	49	8.4
	Single	258	44.2
	Unformalized union	75	12.8
	Widower	5	0.9
Country of residence	Spain	167	28.6
	South America	275	47.1
	Central America and Caribbean Islands	54	9.2
	North America	88	15.1
Profession	Technical careers	78	13.4
	Psychologist	40	6.8
	Educator	16	2.7
	Male nurse	120	20.5
	Sciencehealth student	39	6.7
	Medical	207	35.4
	Other	84	14.4
Provided care to COVID-19 patients	Yes	334	57.2
	No	250	42.8

**Table 2 healthcare-11-00844-t002:** Coping strategies and styles.

Coping Strategies	
Adaptive/helpful coping	Nonadaptive/unhelpful coping
1. Positive reappraisal: Creating a new meaning of the situation by doing something good about the problem.	2. Depressive reaction: Feeling overwhelmed by the situation, being distrustful of oneself, and pessimistic about the results of the problem.
4. Planning: Efforts to change the situation based on an analytical, rational, and experiential approach to the problem.	3. Denial: Lack of acceptance and avoidance of reality, distorting or disfiguring the problem.
5. Acceptance: Acceptance of the lack of personal control over the situation and acquiescence to its consequences and tolerating having unmanageable problems.	6. Cognitive disengagement: Using distracting thinking to avoid focusing on the problem
7. Personal development: Consider the problem as a stimulus and an opportunity for learning and personal growth.	8. Emotional concealment: Efforts to hide personal emotions from others.
10. Elimination of distracting activities: Efforts to stop activities that prevent you from concentrating on understanding and solving the problem.	9. Emotional distancing: Cognitive efforts to suppress the emotional outcomes generated by the situation.
11. Coping constraint: Reduce and defer any management efforts until complete information about the problem is obtained.	12. Coping with suppression: Stopping courses of action for fear that any effort might make things worse or deeming the problem impossible to solve.
13. Problem-solving: Decide and take a direct and reasoned action to manage the problem.	15. Behavioral disengagement: Avoidance of any response or action to solve the problem.
14. Social support for problem-solving: Seeking information, advice, or help from others to solve the problem.	17. Social emotional support: Seeking sympathy and comfort from others for one’s own emotions.
16. Emotional expression: Express to others your own emotional reactions generated by the situation.	18. Palliative response: Avoiding the problem through maladaptive actions taken, in an attempt to feel better (e.g., alcohol).
Coping styles [Strategies included in each style]	
Method Active coping [1, 4, 7, 10, 13 & 16] Passive coping [2, 5, 8, 11, 14 & 17] Avoidance [3, 6, 9, 12, 15 & 18]	
Attention Focused on response [1, 2, 3, 10, 11 & 12] Focused on the problem [4, 5, 6, 13, 14 & 15] Focused on emotion [7, 8, 9, 16, 17 & 18]	
Exercise Cognitive coping [1, 2, 3, 4, 5, 6, 7, 8 & 9] Behavioral coping [10, 11, 12, 13, 14, 15, 16, 17 & 18]	

**Table 3 healthcare-11-00844-t003:** Comparison of anxiety levels due to COVID-19 among health personnel in Latin American countries and Spain.

Anxiety Measures	M (SD)		*t*	Next (2-Sided) *p*-Value
	Spain (n = 167)	Latin America (n = 417)		
Direct anxiety score	34.18	33.52	0.857	0.392 (n.s.)
	(8.514)	(8.439)		
Anxiety index	42.79	42.04	0.781	0.435 (n.s)
	(10.583)	(10.536)		
Somatic symptoms	11.31	11.28	0.083	0.934 (n.s)
	(3.585)	(4.031)		
Cognitive symptoms	7.51	7.04	2.066	0.039 *
	(2.469)	(2.557)		

* = *p* < 0.05; ns = non-significant difference.

**Table 4 healthcare-11-00844-t004:** Differences by sex in the levels of anxiety due to COVID-19 among health personnel.

Anxiety Measures	M (SD)		*t*	Next (2-Sided) *p*-Value
	Women (n = 424)	Men (n = 160)		
Direct anxiety score Anxiety index Somatic symptoms Cognitive symptoms	34.20 (8.407) 42.87 (10.484) 11.52 (3.945) 7.30 (2.550)	32.39 (8.480) 40.63 (10.570) 10.67 (3.741) 6.84 (2.487)	2.311 2.298 2.349 1.966	0.021 (*) 0.022 (*) 0.019 (*) 0.050 (*)

* = *p* < 0.05.

**Table 5 healthcare-11-00844-t005:** Comparison of strategies for coping with COVID-19 between health personnel in Latin American countries and Spain.

Coping Strategies	M (SD)		*t*	Next (2-Sided) *p*-Value
	Spain (n = 167)	Latin America (n = 417)		
Active method Passive method Avoidance method Focus on response Focus on the problem Focus on emotion Cognitive activity Behavioral activity	40.48 (12.213) 31.69 (8.227) 20.85 (8.067) 27.97 (8.300) 32.10 (9.027) 32.95 (7.917) 50.28 (12.802) 42.74 (11.623	38.30 (14.227) 29.37 (10.549) 23.39 (10.056) 27.56 (10.069) 31.27 (11.212) 32.23 (10.991) 49.33 (16.684) 41.73 (15.163)	1.739 2.547 −2.908 0.471 0.851 0.768 0.658 0.779	0.083 (n.s.) 0.010 (**) 0.004 (**) 0.638 (n.s.) 0.395 (n.s.) 0.443 (n.s.) 0.511 (n.s.) 0.436 (n.s.)

** = *p* < 0.01; ns = non-significant difference.

**Table 6 healthcare-11-00844-t006:** Correlations between anxiety and coping.

	Anxiety P. Dir.	Anxiety Index	Somatic Symptoms	Cognitive Symptoms	Active Method	Passive Method	Avoid. Method	Focal Answer	Focal Problem	Focal Emoc.	Active Cond.
Anxiety index	0.999 **										
Somatic symptoms	0.889 **	0.889 **									
Cognitive symptoms	0.834 **	0.834 **	0.695 **								
Active method	−0.122 **	−0.122 **	−0.021	−0.042							
Passive method	−0.018	−0.019	0.041	0.026	0.772 **						
Avoid. method	0.152 **	0.152 **	0.146 **	0.134 **	0.397 **	0.628 **					
Focal answer	0.033	0.033	0.071	0.084 *	0.765 **	0.852 **	0.778 **				
Focal problem	−0.065	−0.065	0.018	−0.013	0.878 **	0.884 **	0.651 **	0820 **			
Focal emoc.	−0.002	−0.003	0.063	0.029	0.838 **	0.883 **	0.679 **	0.798 **	0.843 **		
Activated cognition	−0.054	−0.055	0.025	−0.002	0.878 **	0.868 **	0.697 **	0.877 **	0.919 **	0.903 **	
Active cond.	0.032	0.032	0.080	0.070	0.805 **	0.912 **	0.729 **	0.894 **	0.892 **	0.891 **	0.824 **

**. The correlation is significant at the 0.01 level; *. The correlation is significant at the 0.05 level.

**Table 7 healthcare-11-00844-t007:** Differences in anxiety levels among healthcare personnel depending on whether they cared for patients with COVID-19.

Anxiety Measures	M (SD)		*t*	Next (2-Sided) *p*-Value
	YES Pat. COVID (n = 334)	NO Pat. COVID (n = 250)		
Direct anxiety score Anxiety index Somatic symptoms Cognitive symptoms	34.12 (8.538) 42.74 (10.679) 4.74 (1.857) 10.61 (2.570)	33.15 (8.336) 41.60 (10.352) 4.58 (1.727) 10.54 (3.097)	−1.369 −1.285 −1.065 −0.331	0.172 (n.s.) 0.199 (n.s.) 0.287 (n.s.) 0.741 (n.s.)

ns = non-significant difference.

**Table 8 healthcare-11-00844-t008:** Summary of the hierarchical regression analysis for factors predicting anxiety.

Predictive Factors	*B*	*EE*	β
Model 1			
Avoidance Method	0.134	0.036	0.152 ***
	(*R*^2^ = 0.023) *F* (1. 582) = 13.855 ***		
Model 2			
Avoidance Method	0.326	0.049	0.370 ***
Cognitive Activity	−0.168	0.030	0.312 ***
	(*R*^2^ = 0.073)		
	*F* (2. 581) = 22.933 ***		
Model 3			
Avoidance Method	0.348	0.050	0.395 ***
Cognitive Activity	−0.179	0.030	−0.331 ***
Location (Spain)	−1.716	0.759	−0.092 *
	(*R*^2^ = 0.081)		
	*F* (3. 580) = 17.101 ***		

***. Significant at the 0.001 level; *. Significant at the 0.05 level.

## Data Availability

Data is not available due to privacy restrictions.

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
