# Peer review of "Anxiety Levels and Coping Strategies to Deal with COVID-19: A Cross-Cultural Study among the Spanish and Latin American Healthcare Populations"

_healthcare, 2023, doi:10.3390/healthcare11060844_

Round 1

Reviewer 1 Report

Dear Author, please find my comments and good luck.

WHAT IS THE MAIN QUESTION ADDRESSED BY THE RESEARCH? Currently, the world is facing the threat of the coronavirus pandemic caused by the newly developed and evolving SARS-CoV-2 virus, responsible for COVID-19 disease. Daily worries related to COVID-19 issues are concerning for the psychological maladjustment of the person, who must cope with stress related to various potentially dangerous conditions, with the likelihood of developing depression and/or anxiety symptoms. In this context, the authors aimed to analyze the level of anxiety as well as coping strategies among health personnel in Latin American countries versus Spain.

IS THE TOPIC ORIGINAL OR RELEVANT IN THE FIELD? The subject under study is certainly important, especially in the historical period we are experiencing. The article presents interesting results but, it must be improved, especially for some methodological concerns.

DOES IT ADDRESS A SPECIFIC GAP IN THE FIELD? The authors should make clearer what is the gap in the literature that is filled with this study, and what are the most confounding axiety factors in the post COVID-19 acute emergency.

WHAT SPECIFIC IMPROVEMENTS SHOULD THE AUTHORS CONSIDER REGARDING THE METHODOLOGY? Title: It can be improved reporting the place where the study was conduced, being more attractive to the readers.

Introduction: If the authors plan to conduct a research and would like to explain the problem, they must start by presenting the issue, the paper must be first of all included in the frame of the current emerging priorities raising anxiety (refer to articles with DOI: 10.3390/ijerph191911929), which also include COVID-19. Then they must report the knowledge already existing and that they will consider. Finally, they must show what they want to do and how they want to do it.

Methods:

1. The authors talk about a minimum sample size and the instrument to its calculation, but the minimum sample is not reported and it is not clear what is the reference population: all Spanish and Latino-American HCW? How large is the reference population? Without the numerical identification of the reference population is not clear the validity of the study. A non-representative sample is by its self a non-sense-survey.

2. The selection of the sample raises many criticisms. It is not completely clear how was it selected. How did this method allowed to avoid selection bias. This still requires detailed explanation.

3. Measurements are also not very clearly described. For instance, I am unsure what the alpha values listed refer to. If some numerical scales (interval variables) are present, then the acceptable (internal consistency) reliability, as measured by alpha, does not guarantee that the scales are unidimensional, this should be checked with confirmatory factor analysis.

4. The survey was conducted using a non-standard questionnaire. The use of an unreliable instrument is a serious and irreversible limitation of the study. The fact that the questionnaire construction refers to very old previously used surveys is not sufficient. A validation process must be performed to evaluate the tool.

ARE THE CONCLUSIONS CONSISTENT WITH THE EVIDENCE AND ARGUMENTS PRESENTED AND DO THEY ADDRESS THE MAIN QUESTION POSED? I also suggest expanding. Emphasize the contribution of the study to the literature. The discussion must be updated with the comparison and discussion regarding similar experiences in other international context (see the above mentioned reference). After describing the study results, it would be appropriate to discuss about the strengths and future prospects, which originate from your work, in terms of Public Health. What is the added value provided by the study? It would be useful to examine in depth these aspects, otherwise it seems you have described a mere survey without being forward-looking.

Reviewer 2 Report

I thank the authors for the opportunity to read their article.

The article is well designed and correctly written. Below are some of my comments for authors to consider.

General note: footnotes should be in square brackets

In "Introduction" there are almost no articles from 2022. This is a serious shortcoming because a lot of research has just appeared in 2022. Especially from line 73 it is visible. Please complete this section with the most important research results from 2022. Many of the effects of the pandemic have deferred effects, which means that in 2022 we may see some of them.

Instruments

the authors used the "Zung Anxiety Scale" What was the reason for using such an old tool?

Procedure

Please specify the date of the study.

Was there a question in this study about whether medical staff work or not work in contact with people infected with covid-19? People who have direct contact with patients infected with covid-19 feel a different level of fear

Results

Page 7: „Women showed higher levels of anxiety due to COVID-19 than men.” This opinion is not entirely valid. women tend to be more anxious than men. I would rather emphasize that not because of covid-19 but during covid-19. In addition to the pandemic, there are other factors that affect the results of women

Discussion

Since R2  was small (table 8), what other factors not included in the study, according to the authors, may affect anxiety?

In this part, as in the introduction, there are few references to research conducted in 2022

Round 2

Reviewer 1 Report

the paper was improved and it is now suitable for publication

Reviewer 2 Report

I got acquainted with the second version presented by the authors. Their corrections significantly improved the quality of the article. I congratulate the authors on taking up the topic and encourage them to continue their research